# Investigation of the Mechanical Properties of Reinforced Calcareous Sand Using a Permeable Polyurethane Polymer Adhesive

**DOI:** 10.3390/ma17215277

**Published:** 2024-10-30

**Authors:** Dingfeng Cao, Lei Fan, Rui Huang, Chengchao Guo

**Affiliations:** 1School of Civil Engineering, Sun Yat-sen University, Zhuhai 519082, China; caodf3@mail.sysu.edu.cn (D.C.); guochch25@mail.sysu.edu.cn (C.G.); 2Guangdong Key Laboratory of Marine Civil Engineering, Guangzhou 510275, China; 3Guangdong Research Center for Underground Space Exploitation Technology, Guangzhou 510275, China; 4Southern Institute of Infrastructure Testing and Rehabilitation Technology, Huizhou 516000, China; 5Henan Zhonggong Design & Research Group Co., Ltd., Zhengzhou 450000, China

**Keywords:** calcareous sand, foundation reinforcement, polyurethane polymer, soil deformation, settlement, creep, mechanical properties

## Abstract

Calcareous sand has been widely used as a construction material for offshore projects; however, the problem of foundation settlement caused by particle crushing cannot be ignored. Although many methods for reinforcing calcareous sands have been proposed, they are difficult to apply on-site. In this study, a permeable polyurethane polymer adhesive (PPA) was used to reinforce calcareous sands, and its mechanical properties after reinforcement were investigated through compression creep, direct shear, and triaxial shear tests. The reinforcement mechanism was analyzed using optical microscopy, CT tomography, and mercury intrusion porosimetry. The experimental results indicate that there is a critical time during the compression creep process. Once the critical time is surpassed, creep accelerates again, causing failure of the traditional Burgers and Murayama models. The direct shear strength of the fiber- and geogrid-reinforced calcareous sand reinforced by PPA was approximately nine times greater than that without PPA. The influence of normal stress was not significant when the moisture content was less than 10%, but when the moisture content was more than 10%, the shear strength increased with an increase in vertical normal stress. Strain-softening features can be observed in triaxial shear tests under conditions of low confining pressure, and the relationship between the deviatoric stress and strain can be described using the Duncan–Chang model before softening occurs. The moisture content also has a significant influence on the peak strength and cohesive force but has little influence on the internal friction angle and Poisson’s ratio. This influence is caused by the different PPA structures among the particles. The higher the moisture content, the greater the number of pores left after grouting PPA.

## 1. Introduction

Coral calcareous sand (CCS) is a unique sediment found extensively in tropical regions between 30° north and south latitudes which forms from marine plants and reworked shell fragments after experiencing complex geological and biological processes [1,2,3]. Because of the high transportation cost of traditional quartz river sand, CCS has become the main building material for reclamation projects such as runway foundations, offshore platforms, and marine underground structures. The carbon emission and transportation energy consumptions can reach 0.009 kg and 0.13 MJ when one ton of raw construction materials is shipped per kilometer [4]. Compared to quartz sand, CCS possesses distinctive characteristics such as irregular shape, sharp edges and corners, abundant internal pores, high compressibility, and excellent permeability [5,6,7]. As a result, when CCS is directly used as a building foundation, it can lead to foundation settlement due to factors such as load impact, groundwater level fluctuation, typhoons, and earthquake disturbances, which can further damage the superstructure [4]. Several reinforcement approaches have been proposed to address the vulnerability of coral sand to breakage. These reinforcement approaches can be divided into three types: physical, biological, and chemical grouting [8].

Physical treatments include static pressure, vibration, and fiber (or geogrid) reinforcement [9,10,11,12,13]. Both static pressure and vibration can increase the compressive and shear strengths of CCS foundations by improving the dry density and reducing the porosity, which are mandatory in construction regulations. However, after compression and vibration treatment, the CCS foundation produces significant creep settlement owing to its crushability [14,15,16,17]. To reduce the long-term creep settlement, Wang et al. [18] improved tensile strength of CCS samples by adding polypropylene, carbon, and basalt fibers and reported that the strength was increased by 234.0%, 11.8%, and 100.7%, respectively. The deviatoric stress–strain curves changed from softening to hardening with the addition of fibers [8]. The liquefaction resistance can also be improved by fibers, and a linear relationship has been established to describe the relationship between the liquefaction resistance and loading cycles [19]. However, it is an unavoidable fact that all the physical treatment measures are only applicable prior to the marine constructions and cannot be used during the operation period.

The biological modification commonly refers to two types: microbially induced carbonated precipitation (MICP) and enzymatically induced carbonate precipitation (EICP) [2]. MICP uses urease produced by live ureolytic bacteria to catalyze urea hydrolysis. The released carbonate ions then combine with cation ions under the action of ionic bond attraction, accompanied by calcium carbonate precipitation [20]. Therefore, the MICP process is heavily dependent on the urease produced by live ureolytic bacteria such as Bacillus, Sporosarcina, Costridium, and Desulfotomaculum [20]. While these bacteria can be effectively cultivated in the laboratory and have been proven to be active, it is currently unclear whether they can survive or remain highly active on the artificially filled islands in the South China Sea. Bacteria survival is influenced by many factors, including temperature, humidity, salinity, trace elements, organic matter content, and the presence of other bacterial flora categories. In addition, Zhang et al. [21] revealed that calcium carbonate crystals could not grow on negatively charged bacterial surfaces at the micron scale, and the previous MICP theory of bacterial cells acting as nucleation sites was oversimplified.

In EICP, urease is extracted from plants, such as beans, melons, and squash, or from live ureolytic bacteria using sonication approaches [20]. Compared with MICP, EICP is more suitable for clay soils with lush vegetation. Therefore, the treatment depth of EICP is limited by the plant roots, which limits the treatment of foundation creep in the South China Sea islands at large depths. Another significant shortcoming of MICP and EICP is their low efficiency and slow processing times. When engineering problems arise in island projects (e.g., airport runway drainage issues, uneven foundation settlement, and slope instability), there is an urgent need for rapid repairs. However, MICP and EICP require several months, or even years, to address these issues. The excessive processing period will cause further evolution of the disaster and seriously restrict the normal use of the structures.

Chemical reinforcement methods have been proposed to overcome the shortcomings of the physical and biological methods for the rapid repair of foundations. The materials used include traditional cement and lime, as well as newly developed polyurethane polymers. The advantages of cement (or lime) are low cost and high strength, whereas the disadvantage is that the treated foundation is highly brittle, stiff, and prone to cracking and damage under dynamic loads or impact loads [22]. The chemical materials used in construction foundation reinforcement include polyethylene terephthalate, polyurethane foam adhesive (PFA), and polyurethane polymer adhesive (PPA) [23]. PFA includes two components, usually defined as A and B. A high-strength foamed solid is generated after liquid A and B components are mixed in a certain proportion. Tao et al. [22] has successfully used PFA to strengthen CCS and evaluated its enhanced strength and stiffness under the conditions of PFA contents of 3%, 4.5%, 6%, and 7.5% through a series of triaxial tests. During the sample preparation process, CCS and liquid B were mixed and thoroughly stirred, after which liquid A was added and the mixture was stirred. However, it is difficult to apply this mixing and stirring method on site, particularly for building foundations. In general, liquids A and B have high viscosities and are difficult to diffuse directly in fine media, such as CCS. The polymer slurry splits the weak surface of the soil layers under pressure, eventually forming a sheet-like foam solid [24,25]. Compared to PFA, PPA has a higher permeability and lower expansion force and is more suitable for porous media with small pore sizes [23,26].

To fill this gap, a new grouting equipment was designed to prepare soil samples with PPA for geotechnical tests and replace the traditional mixture-stirring method. Compression, direct shear, and compression triaxial tests were conducted. The microstructure of CCS reinforced with PPA was studied using optical microscopy, CT scanning, and mercury intrusion porosimetry.

## 2. Materials and Methods

### 2.1. Calcareous Sand and Permeable Polymer Adhesive Materials

The CCS used in this study was collected from the South China Sea near the Xisha Islands, and its particle size distribution is shown in Figure 1. The effective grain size (*d*_10_), median grain size (*d*_30_), and control grain size (*d*_60_) were 0.044, 0.102, and 0.280 mm, respectively. The uniformity coefficient *C*_u_ (*d*_60_/*d*_10_) and the curvature coefficient *C*_c_ ($d_{30}^{2}/d_{10}d_{60}$) were 6.364 and 0.844, respectively. According to the unified soil classification system (USCS) of the United States of America, *C*_u_ > 6 indicates well-graded sand [27]. The liquid limit ($\omega_{L}$) and plastic limits ($\omega_{p}$) were 19.8% and 15.3%. The chemical components (without considering oxygen and carbon elements) obtained through X-Ray Fluorescence are listed in Table 1. Calcium accounted for 93.888%, mainly in the form of calcium carbonate.

The PPA used in this study was a two-component (referred to as A and B) adhesive polyurethane with environmental friendliness, safety, and minimal odor. The main chemical components of PPA are listed in Table 2. The main composition of A is diphenylmethane diisocyanate. The compositions of B include polyether or polyester polyols, tris(1-chloro-ethylpropyl) phosphate, and additive. When A and B are mixed, two main separate chemical-independent reactions occur, as shown in Figure 2. The chemical yields mainly include carbanmate and isocyanurate. The first is the gel reaction, in which the hydroxyl group in the polymerized polyol reacts with the isocyanate group to form a polymer containing a urethane segment, as shown in Figure 2a. The urethane groups have a high cohesive energy and are characterized as hard segments in the polymer. Because polyol has a carbon chain, its deflection effect makes it a soft segment of the polymer. Therefore, polyurethane is a segmented copolymer composed of hard and soft segments. The urethane group imparts polyurethane with excellent compressive strength, whereas the ether bond segment imparts polyurethane with excellent toughness. The other is a trimerization reaction, as shown in Figure 2b. Under the action of a trimerization catalyst, isocyanate forms a six-membered heterocyclic structure composed of carbon and nitrogen atoms called isocyanurate. This structure exhibits excellent thermal stability and mechanical strength, which can further improve the compressive strength and heat stability of the material. Some organic metal compounds and nitrogen element compounds can be used as trimerization catalysts for isocyanates. The commonly used tertiary amine trimerization catalysts include N-N-dimethylcyclohexylamine and N-N′-diethylpiperazine. When manufacturing isocyanurate, in order to avoid the formation of useless substances with excessive polymerization due to unlimited reaction, a small amount of inhibitor is generally added to control the degree of trimerization such as phosphoric acid, dimethyl sulfate, etc.

### 2.2. Sample Preparation

Two types of cylindrical samples were prepared for the direct shear and triaxial tests. The first has a diameter of 61.8 mm and a height of 20 mm and is used for compression and direct shear tests. The second had a diameter of 38 mm and a height of 76 mm for the triaxial shear tests. Table 3 lists the moisture contents, dry densities, and stress conditions used in the tests. Soil samples were prepared using the static pressure method. The CCS was dried in an oven at 108 °C for 24 h, and then corresponding water was added. After water addition, the soil samples were placed in sealed plastic bags for 8 h to allow the water to be evenly distributed. The CCS with added water was then placed into the molds and compacted with a hydraulic jack. Unlike traditional molds, the molds used in this study have a grouting function, and the polymer can be injected into the internal soil using a grouting gun, as shown in Figure 3.

The sample preparation mold consisted of two upper and lower circular bases and a cylindrical ring knife. Soil samples were wrapped in an internal space. The upper and lower bases are connected using screws. A concave groove was sculptured at the base, which prevented the ring knife from moving outward under the soil pressure. A removable hoop was placed in the middle of the ring cutter to prevent circumferential deformation. A grouting hole was placed in the lower base and connected to the grouting gun through a flange. Two exhaust pipes were installed in the upper base and connected to a vacuum pump. Gauze was placed between the base and internal soil to prevent sand particles from entering the grouting and exhaust pipes. The grouting gun mainly included a slurry container, a pressure device, and a mixer. Slurries A and B were squeezed into the soil under high pressure after mixing in a specific ratio, with the air discharged through the exhaust pipes. As the slurry diffuses slowly in the fine-grained CCS, two vacuum pumps are connected to the exhaust popes, generating a negative-pressure zone to promote the slurry diffusion rate. When the slurry flows out of the exhaust pipes, it can be considered that the air has been drained, and the entire grouting process can be completed. It should be emphasized that the mold must be maintained vertical during the grouting process, and the upper and lower base positions cannot be reversed. If the slurry is injected from the top, it seeps rapidly along weak surfaces such as boundaries, forming a spindle instead of filling the entire soil, making it impossible to prepare cylindrical samples. The soil-filling procedures in the mold were the same as in traditional geotechnical tests and could be completed by static pressure or layer-by-layer compaction (the static pressure method was used in this study). Grouting began after the soil samples were filled. The mold has two specifications: one with an inner diameter of 38 mm and a height of 78 mm, and the cylinder formed in it can be directly used for triaxial tests, as shown in Figure 4a. The other had an inner diameter of 61.8 mm and a height of 100 mm, and the formed cylinder needed to be manually cut into shorter samples with a height of 20 mm for direct shear and compression creep tests.

### 2.3. Test Equipment and Procedures

#### 2.3.1. Compression Creep Test

The compression creep test was performed using a one-dimensional high-pressure oedometer, as shown in Figure 5a. The load range of the oedometer was 12.5–4000 kPa. The hydraulic conductivity of the permeable board was larger than 1 × 10^−5^ m/s. The moisture content of the soil sample was 10%, and the total test period was 22 d. The data were analyzed every 24 h. The vertical loads were set to 100, 200, and 400 kPa. The pressurized cover plate was wrapped with wet cotton gauze to which water was added regularly to ensure a moist environment and avoid evaporation of the soil sample.

#### 2.3.2. Direct Shear Test

A quadruple direct-shear device apparatus was used, as shown in Figure 5b. A total of 120 soil samples were collected, with 24 samples for each moisture content. Six vertical loads were applied: 100, 200, 300, 400, 600, and 800 kPa (Table 3). Prior to the test, a fixed pin was inserted into the sample container to prevent disturbances. Permeable stones and a filter paper were placed on the upper and lower surfaces of the soil samples. After the sample installation was finished, the fixed pin was quickly pulled out at the beginning of soil shearing. The shear rate was controlled at 0.8 mm/min. The horizontal force and displacement were recorded every 5 s during the testing. If there was a peak force value, the test was continued until the shearing displacement reached 4 mm. If no peak value was observed, the test was terminated when the horizontal displacement reached 6 mm.

#### 2.3.3. Triaxial Shear Test

An unsaturated soil triaxial device was used for the triaxial tests. The test type was a triaxial consolidation drainage shear. A rubber film was wrapped around each soil sample column and filter paper was placed at each end. The shear rate was 0.009%/min until the stress peak or the axial strain reached 10%.

#### 2.3.4. Microstructural Analysis

An optical microscope, CT scan, and mercury intrusion test were conducted to investigate the solid particle connections, distribution of polymers, and pore features, respectively. The model of the optical microscope used in the experiment was an ANDONSTAR-3800-4K (produce by ProjectorScreen, New Jersey, America) with a maximum magnification of 135×, 38 million effective pixels, and a frame rate of 60 frames/s. Three soil samples with water contents of 0%, 5%, and 10% were used to determine the influence of moisture content. The CT equipment was produced by Comet Yxlon (Hamburg, Germany), and the model used was the FF35. The mercury intrusion porosimetry test was finished using the MacAutoPore V 9600 (produce by Micromeritics Instrument Corporation, Georgia, America) with a pressure range of 0.5–33,000 psi and a pore size measurement range from 350 μm to 5 nm. Mercury enables the invasion of pores of different sizes under different pressures, and the larger the external pressure, the smaller the pore radius that the mercury can invade. Therefore, the volume of mercury entering the pores at different external pressures was measured to determine the corresponding pore volume.

The relationship between the intrusion pressure $P_{L}$ and the pore diameter $D_{P}$ satisfies the following equation:(1)$$P_{L}-P_{G}=\frac{-4\sigma \cos\theta }{D_{P}}$$
where *P* is the intrusion pressure (kPa), $P_{G}$ is the initial gas pressure (kPa), $D_{P}$ is the pore diameter (μm), and *θ* represent the contact angle between the solid and the mercury [28]. As the test was performed under vacuum, the initial gas pressure $P_{G}$ was equal to zero. At the room temperature of 20 °C under vacuum conditions, the surface tension of mercury *γ* is 480 × 10^−3^ N/m, and the contact angle *θ* can be taken as 140°. Therefore, Equation (1) can be simplified as
(2)$$D_{P}=\frac{1470}{P_{L}}$$

According to the fractal theory proposed by Mandelbrot, the backbone fractal dimension $D_{\mathrm{backbone}}$ and percolation fractal dimension $D_{\mathrm{percolation}}$ can be used to quantitatively evaluate the pore distribution in porous media [29]. $D_{\mathrm{Backbone}}$ and $D_{\mathrm{Percolation}}$ are defined as:(3)$$D_{\mathrm{Backbone}}=3-\frac{\log S_{\mathrm{Hg}}}{\log(\frac{P_{i}}{P_{\min}})},\;P_{i}<P_{\mathrm{Displacement}}$$
(4)$$D_{\mathrm{Percolation}}=3-\frac{\log S_{\mathrm{Hg}}}{\log(\frac{P_{i}}{P_{\mathrm{Displacement}}})},\;P_{i}>P_{\mathrm{Displacement}}$$
where $P_{i}$ denotes the intrusion pressure corresponding to the pore diameter, $P_{\min}$ is the minimum intrusion pressure, and $P_{\mathrm{Displacement}}$ is the critical intrusion pressure. $S_{\mathrm{Hg}}$ is defined as:(5)$$S_{\mathrm{Hg}}=\frac{V_{\mathrm{Cumulate}}(D<D_{i})}{V_{\mathrm{Cumulate}}}=\frac{D_{i}^{\left(3-D_{v}\right)}-D_{\mathrm{Min}}^{\left(3-D_{v}\right)}}{D_{\mathrm{Max}}^{\left(3-D_{v}\right)}-D_{\mathrm{Min}}^{\left(3-D_{v}\right)}}$$
where $V_{\mathrm{Cumulate}}(D<D_{i})$ is the accumulated mercury intrusion volume when the pore diameter is less than $D_{i}$; $V_{\mathrm{Cumulate}}$ is the total accumulated mercury intrusion volume; $D_{\mathrm{Max}}$ is the maximum pore diameter; $D_{\mathrm{Min}}$ is the minimum pore diameter; and $D_{V}$ is the fractal dimension of the cumulative pore volume, which can be obtained from the mercury intrusion test.
(6)$$\frac{dV}{dD}=\frac{\pi }{6}D^{\left(2-D_{V}\right)}$$

## 3. Results and Discussion

### 3.1. Compression Creep Development with Elapsed Time

Figure 6 shows the relationship between the vertical creep strain and elapsed time. In the creep mechanism of quartz sand, as the pore water is discharged, the effective stress increases, and the particles are redistributed. Compared with quartz stone, CCS has a dual-pore structure; that is, in addition to the pores between particles, there are also pores inside the particles, making them easy to crush [30]. Therefore, the creep of CCS under a load includes particle redistribution and fragmentation [15]. Common models describing the creep of CCS include the Murayama and Burgers models [31,32]. The root mean square deviations (RMSDs) and *R*^2^ of the Murayama and Burgers models are listed in Table 4.

The Burgers model was established based on a Kelvin body and the Maxwell model connected in series [32]:(7)$$\varepsilon =\frac{\sigma }{E_{0}}+\frac{\sigma }{E_{1}}\left(1-e^{\frac{E_{1}}{\eta_{1}}}\right)+\frac{\sigma }{\eta_{2}}t$$
where ε is the total strain, σ is the stress (kPa), t is the creep time (min), $E_{0}$ is the elastic modulus of the Hooke body (MPa), $E_{1}$ is the elastic modulus of the Kelvin body (MPa), $\eta_{1}$ is the viscosity coefficient of the Hooke body (kPa), and $\eta_{2}$ is the viscosity coefficient of Kelvin (kPa). The creep strain development is separately discussed in the Murayama model [31]:(8)$$\varepsilon =\left\{\begin{array}{l} \frac{\sigma }{E_{0}},\;\sigma \leq \sigma_{s}\\ \frac{\sigma }{E_{0}}+\frac{\sigma -\sigma_{s}}{E_{1}}\left(1-e^{-\frac{E_{1}t}{\eta_{1}}}\right),\;\sigma >\sigma_{s} \end{array}\right.$$
where $\sigma_{s}$ denotes the yield stress (MPa). For a specific soil, only ε and t are variables, and the other parameters can be fitted using the least-squares method, as shown by the curves in Figure 6. Table 4 lists the *R*^2^ values of the fitted curves for the Burgers and Murayama models.

It can be inferred from Figure 6 that there are time thresholds (*t*_t_) in the strain–time curves. Before *t*_t_, the measured and predicted values of the Burgers and Murayama models were consistent, and the compressive deformation rate changed from fast to slow. After the *t*_t_, the measured deformation data began to increase again, and neither the Burger nor Murayama models reflected this restarted deformation. The main reason for the deviation between the measured and model-predicted values after the critical values was that the damage mechanism of CCS after grouting was completely different from that of pure sand. After the calcareous sand without grouting was compressed, the fine particles gradually filled the pores around the coarse particles [33]. After the irregular convex parts of the calcareous sand particles were broken, the contact area of the adjacent particles increased, and fewer particles could be broken. The gradual increase in the particle contact area was reflected in the gradual decrease in the compressive creep rate [14,15,33]. After PPA was injected into CCS, it wrapped around the particles and completely changed the way the particles contacted each other. The strength, toughness, and cohesion can be significantly improved by grout [22]. Under the long-term action of loads, the polymer produces cutting failure, resulting in intensified creep deformation [34]. Figure 6 also shows that the values predicted by the Burgers and Murayama models were not significantly different. The Murayama model was slightly more accurate than the Burgers model, and all the *R*^2^ are larger than 0.985.

### 3.2. Direct Shear Strength Analysis

Figure 7 shows that the shear strength of the PPA-reinforced CCS increased with increasing moisture content (w). When w = 0%, the direct shear automatically stopped before the shear strain reached 1.5%. When the w was 5%, the test stopped when the shear strain reached 3.5% (the shear stress that the device could provide had reached its maximum value). When the w was larger than 10%, the shear strength decreased. This is because the pore water makes it difficult for the slurry to enter the pores, resulting in a decrease in strength [13]. The influence of vertical loads was not significant when the w was less than 10%, and this was because when the moisture content was low, the water was mainly wrapped around the solid particles and had not yet formed a connecting water bridge between the particles, which had little effect on the diffusion of the slurry [35].

Under higher-moisture-content conditions (w > 10%), the shear strength increased with an increase in the vertical normal stress. This was because there was less slurry in CCS with a high moisture content, and the solid particles were not firmly adhered; therefore, the sand particles were prone to move under the action of an external force. The increase in normal stress causes the sand particles to be more closely arranged, the particle surface friction increases, and the shear strength increases accordingly [36]. The peak of the shear stress, or the stress corresponding to a displacement of 6 mm, was suggested by considering the shear strength. The shear displacement corresponding to a shear stress of 6 mm was considered as the shear strength of the composite at different moisture contents. The linear Mohr–Coulomb strength theory has been suggested to fit the relationship between shearing and normal stress [36]:(9)τ=c+σtanφ
where c is the cohesion (kPa), and φ is the internal friction angle (°). The envelope lines of the Mohr–Coulomb criterion strengths are shown in Figure 8.

According to the operating specifications, the strength cannot be calculated from the direct shear curves of soil samples with moisture contents of 0 and 5% because the stress-displacement curve does not have a peak and the total displacement reaches a total level of 6 mm. However, this phenomenon indicates that the shear strength is very high under these moisture content conditions, and the strength value should be further determined using larger-scale equipment. The strength of the soil with a moisture of 20% was close to that of the fiber-reinforced CCS, and the strength of the grouted samples with other moisture contents was much higher than that of fiber-reinforced soil [13]. Compared with the maximum direct shearing strength of the geogrid-reinforced CCS, the maximum value of the grouted sand in this study increased by nine times [36].

### 3.3. Triaxial Shear Strength Analysis

The Kondner model established by Hooke’s law is the most common model for describing the relationship between $\sigma_{1}-\sigma_{3}$ and $\varepsilon_{1}$ and is expressed as [37,38]
(10)$$\sigma_{1}-\sigma_{3}=\frac{\varepsilon_{1}}{a+b\varepsilon_{1}}$$
where $\sigma_{1}$ and $\sigma_{3}$ denote the maximum and minimum principal stresses (kPa), respectively. *a* and *b* represent the experimental constants, and $\varepsilon_{1}$ represents the axial strain. The tangential modulus is defined as follows:(11)$$E_{t}=\frac{\partial (\sigma_{1}-\sigma_{3})}{\partial \varepsilon_{1}}=\frac{a}{{\left(a+b\varepsilon_{1}\right)}^{2}}$$

According to the Mohr–Coulomb limit balance theory, the relationship between $\sigma_{1}-\sigma_{3}$, c, and φ can be written as
(12)$${\left(\sigma_{1}-\sigma_{3}\right)}_{f}=\frac{2c\cos\varphi +2\sigma_{3}\sin\varphi }{1-\sin\varphi }$$

Janbu proposed that the initial tangent deformation modulus of the soil, $E_{i}$, is related to the confining pressure $\sigma_{3}$, which can be expressed as follows [38]:(13)$$E_{i}=Kp_{a}{\left(\frac{\sigma_{3}}{p_{a}}\right)}^{n}$$
where *K* and *n* are constants determined by the soil type and represent the intercept and slope of $\mathrm{lg}\left(E_{i}/p_{a}\right)$ and $\mathrm{lg}\left(\sigma_{3}/p_{a}\right)$, respectively. $p_{a}$ is the atmospheric pressure with a magnitude of 101.4 kPa. The relationship between $E_{t}$ and $E_{i}$ is defined as
(14)$$E_{t}=E_{i}{\left(1-R_{f}S_{L}\right)}^{2}$$

Because the axial strain $\varepsilon_{1}$ cannot increase infinitely, the failure ratio $R_{f}$ is defined by relating the failure stress ${\left(\sigma_{1}-\sigma_{3}\right)}_{f}$ to the ultimate stress ${\left(\sigma_{1}-\sigma_{3}\right)}_{\mathrm{ult}}$
(15)$$R_{f}=\frac{{\left(\sigma_{1}-\sigma_{3}\right)}_{f}}{{\left(\sigma_{1}-\sigma_{3}\right)}_{\mathrm{ult}}}$$

And $S_{L}$ is defined as:(16)$$S_{L}=\frac{\left(\sigma_{1}-\sigma_{3}\right)}{{\left(\sigma_{1}-\sigma_{3}\right)}_{f}}$$

Duncan and Chang suggested using $S_{L}=70\%$ and $S_{L}=95\%$ to calculate parameters *a* and *b*, and Equation (17) can then be obtained, which is also called the Duncan–Chang model [39].
(17)$${\left(\sigma_{1}-\sigma_{3}\right)}_{\mathrm{ult}}=\frac{{\left(\varepsilon_{1}\right)}_{95\%}-{\left(\varepsilon_{1}\right)}_{70\%}}{{\left(\frac{\varepsilon_{1}}{\sigma_{1}-\sigma_{3}}\right)}_{95\%}-{\left(\frac{\varepsilon_{1}}{\sigma_{1}-\sigma_{3}}\right)}_{70\%}}$$

The cohesive force *c* and internal friction angle φ for different moisture contents were obtained by plotting the Mohr stress circle and shear strength envelopes. The cohesive force *c* decreased with increasing moisture content because the polyurethane polymer strengthened the soil particle skeleton and filled the internal pores, as shown in Figure 9. The effect of the moisture content ω on the internal friction angle φ was not significant. Therefore, it can be inferred that the moisture content mainly affected the cohesive force and had little influence on the internal friction angle. Wu et al. [40] reported that as ω increased, the increase in the internal friction angle was less than 1°, and the total change was less than 3°. The cohesion of sand in classical soil mechanics can be neglected; however, the presence of water in the internal pores can produce matric suction, resulting in an increase in the apparent cohesion. This apparent cohesion gradually decreased and disappeared with increasing ω when free water formed [40].

Figure 10 shows the stress–strain curves of coral sand with different water contents reinforced by grouting with PPA. The stress–strain curves of calcareous sand can be divided into strain-hardening and strain-softening types [41]. When calcareous sand is in a loose state, the stress–strain relationship exhibits strain-softening characteristics, whereas in a dense state, it exhibits strain-hardening characteristics. The stress–strain relationship was primarily determined by the confining pressure. As shown in Figure 10, the sand samples reinforced by PPA exhibited stress peaks and obvious strain-softening characteristics under low confining pressures ($\sigma_{3}$ = 100 or 200 kPa), whereas those under high confining pressure conditions (400 kPa) exhibited strain-hardening features. This change from softening to hardening with confinement agrees well with the observations reported in [42]. Under the same moisture content conditions, the higher the confining pressure, the higher the peak strength. The main factors affecting the stress–strain curve characteristics are the particle structure, dry density, and strength of the polymer. After being subjected to external loads, the CCS was compacted, its porosity was reduced, and the friction between particles increased, resulting in an increase in the internal friction angle [43]. When the deformation accumulated to a certain state, the convex part of the calcareous sand underwent shear failure, and the polymer connectors between the particles experienced cutting failure, resulting in a stress-softening phenomenon in which the stress decreased [34]. It can also be observed from Figure 10 that the higher the moisture content, the lower the peak intensity. This is because the more water there is, the more pores are occupied by water, and less slurry can be injected into the soil. Because the Duncan–Chang model cannot reflect the strain-softening properties, only the data that had not undergone softening were selected for fitting, as shown by the solid curves in Figure 10. The *R*^2^ values of the fitted curves are listed in Table 5. All values of *R*^2^ were greater than 0.93, indicating that the relationship between $\sigma_{1}-\sigma_{3}$ and $\varepsilon_{1}$ can be effectively described by the Duncan–Chang model before softening occurs.

Figure 11 shows the longitudinal strain ($\varepsilon_{l}$) and transverse strain ($\varepsilon_{t}$) of CCS reinforced by PPA; the slope is usually defined as the Poisson ratio (μ). Table 6 shows the μ and *R*^2^ of the linear regression between $\varepsilon_{t}$ and $\varepsilon_{l}$, which were all greater than 0.895, indicating that the Poisson ratio remained unchanged during the process of deformation. The main factors affecting the μ of sand are its relative density, void ratio, and particle size distribution [44]. In order to quantitatively describe the influence of void ratio and confining pressure on μ, a variation function (*G*_max_) was defined in reference [44].

### 3.4. Microstructural Mechanism Analysis of the PPA Reinforced Calcareous Sand

Figure 12 shows optical microscope images of the CCS before and after reinforcement with PPA. When w was 0, the pores were evenly distributed before grouting and almost all the pores were filled with PPA after grouting. When w = 5%, particles have the characteristics of agglomerates before grouting. Several fine particles adhered to each other under the action of capillary force to form an agglomerate, and the entire agglomerate filled the space formed by the large particles. Many small pores, which were occupied by water before grouting, were evenly distributed after grouting and oven-drying. When w is 10% before grouting, agglomerates can be more clearly observed, and a unique internal locking structure in CCS can be observed. Because the surface of CCS particles is uneven, the convex parts of the particles are embedded into the grooves of adjacent particles, thereby limiting the sliding or movement of the particles and resulting in a larger interparticle pore volume [16]. The pore volume directly determines the permeability coefficient. It was reported that the permeability coefficient is positively related to pore volume because the larger the pore volume, the larger the seepage channel and the lower the seepage resistance [45].

The forces between the sand particles can be divided into strong and weak forces [46]. Once long sand particles (spherical CCS particles with smooth surfaces are rare) were stacked on top of each other to form columnar structures. The upper load is mainly borne by the cylinder; therefore, the force acting on the cylinder is also called a strong force. The uncoordinated deformation of the different sand columns produces mutual friction, which is defined as a weak force [46]. The sand–column structure increased the number of voids in the soil. The CCS with an *w* = 10% left significantly larger voids after grouting and oven-drying. From the perspective of engineering foundation treatment, the drier the CCS, the more PPA that can be injected and the better the effect. However, sand cannot be completely dry; therefore, its natural moisture content must be accurately tested before grouting. The Young’s modulus of CCS without grouting can be described by Hertz model which does not account for adhesion forces. Two assumptions are proposed for this model. The first assumption is that the nanoindenter is a perfect sphere that causes a perpendicular penetration in a perfectly planar surface, and the second assumption is that the strain–stress relation is linear by satisfying Hooke’s law. Thus, the Young’s modulus (*E*) can be expressed as [47]:
(18)$$E=\frac{3FR}{4a^{3}}$$
where *F* is the load force (kN), *R* is the sphere radius of the nanoindenter (mm), and *a* is the contact radius between both surfaces. After grouting PPT, strong adhesive force exists among particles, and the Johnson, Kendall, and Roberts (JKR) model considers the short-range forces between surface contact area and external sample surface [47]. The JKR model is used for the occasion with large spherical indenters exposing larger contact areas with larger adhesion events, which can be described as [47]:(19)$$F=\frac{4Ea^{3}}{3R}-2\sqrt{2\pi Ea}$$
where ω is the required energy to separate a unitary area of both surfaces (J).

Three particles with different shapes (*w* = 5%) were selected to investigate connections between particles and PPA using CT scanning technology, as shown in Figure 13. It can be clearly observed that the particle surface was uneven before grouting, and the depressed parts were filled with PPA after grouting. When there was insufficient slurry, the distribution of PPA on the particle surface was not continuous and mainly filled the low-lying areas. When more PPA was supplied and low-lying areas were filled, the PPA connected to each other to form the whole body. The filling and pasting characteristics of PPA are completely different from those of commonly used expanded PFA [23].

Figure 14 shows the relationship between cumulative mercury intrusion volume, increment intrusion volume and pore diameter obtained using mercury intrusion porosimetry method (MIP). The cumulative mercury intrusion volume increased with a decrease in pore diameter because pore diameter has a negative correlation with mercury pressure. As the mercury pressure increased, the pore diameter decreased and more mercury intruded. The final value of the cumulative mercury intrusion volume is the sum of all pore volumes of the sand sample. The cumulative mercury intrusion volume curve can be divided into four stages according to the intrusion rate, as shown in Figure 14a. During stage AB, the intrusion volume increases rapidly, and mercury mainly enters the large pores between the particles. During stage BC, the intrusion volume remains almost unchanged because the energy consumed in this stage is mainly used to compress the soil particles. During stage CD, the intrusion volume slowly increases again, indicating that Hg begins to intrude into the internal pores of the particles. In state DE, the intrusion volume no longer increases, and the cumulative intrusion volume is equivalent to the total pore volume of the sample. As shown in Figure 14b, the pore diameter between particles ranges from 70 to 200 um, and that of the internal pore in the particles ranges from 70 to 120 nm. Additional inferred pore information is presented in Table 7. The median pore diameter volume, V, is defined as the corresponding pore diameter when the volume of mercury fills half the volume of the total pores [48]. The median pore diameter area, A, was defined as the diameter accumulated surface area of 50% [49]. The larger the difference between V and A, the greater the number of types of pores [49].

## 4. Conclusions

In this study, a permeable PPA was used to reinforce calcareous sand, and new grouting equipment was designed to prepare soil samples with PPA for geotechnical tests to replace the traditional mixture-stirring method. The compressive and shearing performances were evaluated using one-dimensional direct shear and triaxial shear tests. The reinforcement mechanism was revealed using optical microscopy, CT tomography, and mercury intrusion porosimetry. The main conclusions are as follows.

(1)There are time thresholds (*t*_t_) in the strain–time curves. Before *t*_t_, the measured and predicted values of the Burgers and Murayama models were consistent, and the compressive deformation rate changed from fast to slow. After the *t*_t_, the measured deformation data began to increase again, and neither the Burgers nor Murayama models reflected this restarted deformation. The deviation after *t*_t_ was caused by the crushing of the calcareous sand particles and cutting failure of the PPA. The larger the confinement pressure, the later *t*_t_ occurs.(2)Compared with the direct shear strength of fiber- and geogrid-reinforced calcareous sands, that reinforced by PPA is approximately nine times greater, which has great practical engineering value. The moisture content (*w*) and vertical normal stress play important roles in determining the direct shear strength of the PPA-reinforced calcareous sand. The influence of normal stress is not significant when the *w* is less than 10% because when the moisture content is low, the water is mainly wrapped around the solid particles and has not yet formed a connecting water bridge between the particles, which has little effect on the diffusion of the PPA. Under higher moisture content conditions (*w* is larger than 10%), the shear strength increased with the increase in vertical normal stress.(3)Strain-softening features were observed in the relationship between $\sigma_{1}-\sigma_{3}$ and $\varepsilon_{1}$ under low confining conditions (100 and 200 kPa). The greater the confining pressure, the higher the peak strength. Before softening occurs, the relationship between $\sigma_{1}-\sigma_{3}$ and $\varepsilon_{1}$ can be effectively described by the classic Duncan–Chang model, with *R*^2^ being greater than 0.93. The higher the moisture content, the lower the peak strength. The moisture content significantly affects the cohesive force but has little influence on the internal friction angle and Poisson’s ratio.(4)The distribution of PPA and pores was determined by the moisture content. The higher the moisture content, the larger the number of pores formed after grouting with PPA. When *w* is 0, almost all the pores can be filled with PPA. When *w* is 5%, evenly distributed small pores can be observed, and when *w* is 10%, interconnected macropores can be observed. Therefore, in actual projects, groundwater must be pumped out before grouting the PPA into a calcareous sand foundation.

In actual engineering applications, further research on grouting methods is needed when using PPA to reinforce calcareous sand, such as the relationship between grouting pressure and solidification time, effective grouting radius, PPA-CCS mixed-body morphology, and grouting pipe layout spacing. When the slurry solidifies too slowly, it may flow along the water flow channel, while when the slurry solidifies too quickly, it may be difficult to form an intact solid, resulting in failure in the foundation treatment.

## Figures and Tables

**Figure 1 materials-17-05277-f001:**
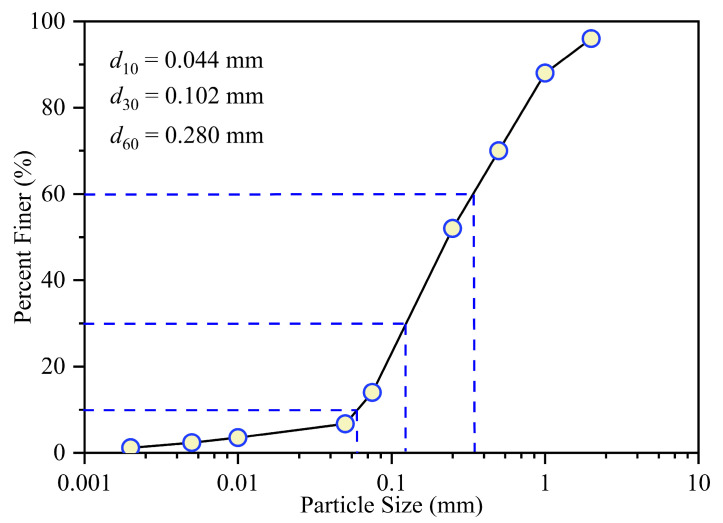
Particle size distribution of calcareous sand.

**Figure 2 materials-17-05277-f002:**
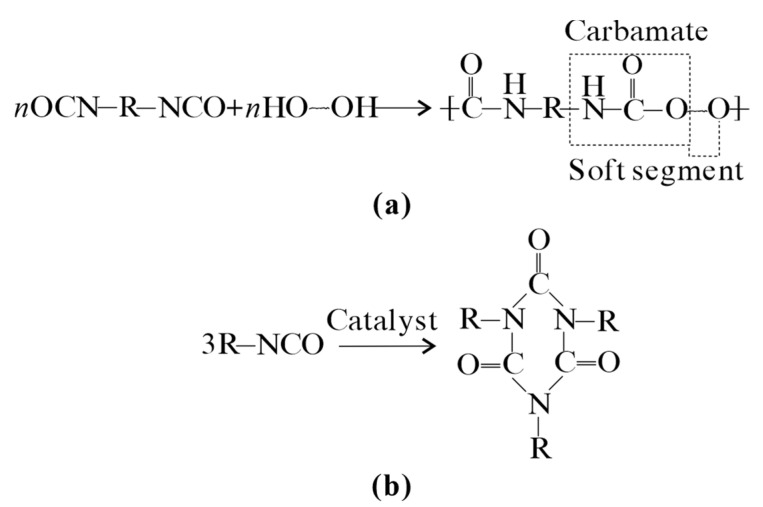
Chemical mechanism of calcareous sand reinforcement using PPA: (**a**) gel reaction and (**b**) trimrization reaction of isocyanurate.

**Figure 3 materials-17-05277-f003:**
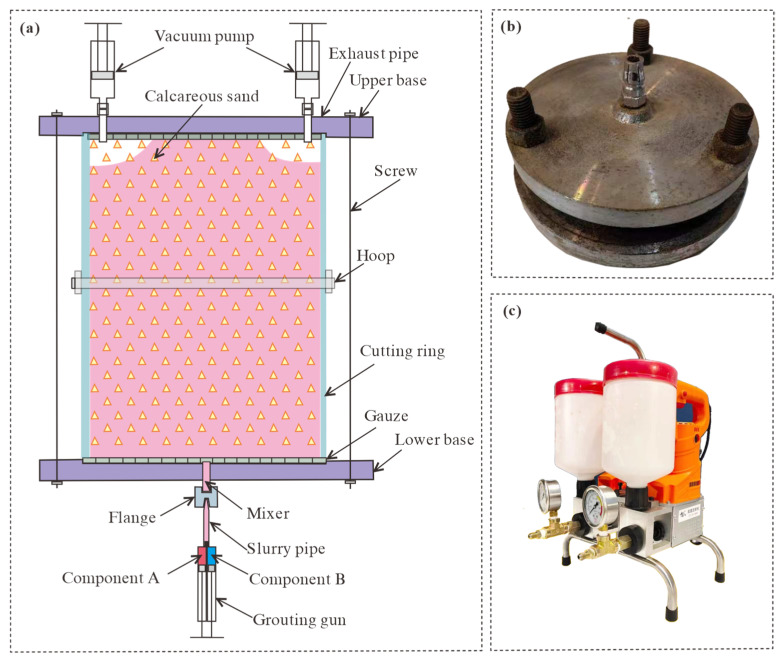
Soil sample preparation setup and grouting system: (**a**) device diagram, (**b**) photo of soil sample making mold, and (**c**) photo of grouting device.

**Figure 4 materials-17-05277-f004:**
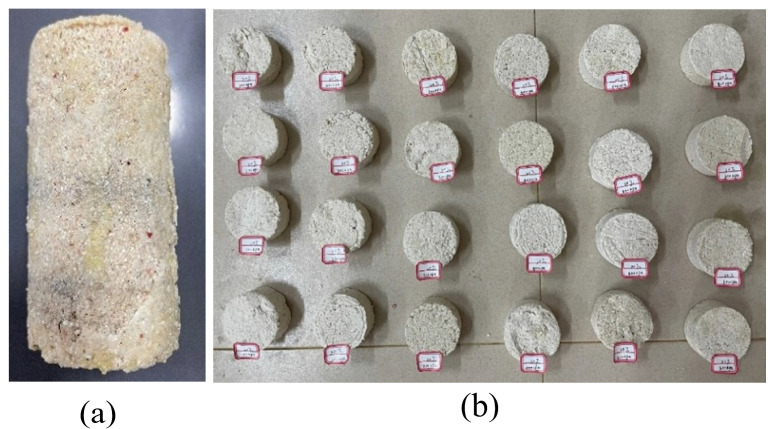
Photos of calcareous sand cylinder samples: (**a**) for triaxial test and (**b**) for direct shear and compression creep tests.

**Figure 5 materials-17-05277-f005:**
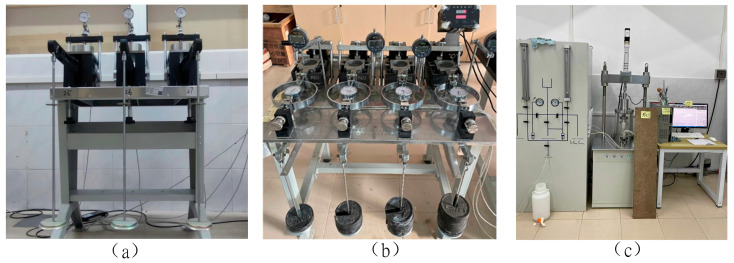
Photos of tests setup: (**a**) one-dimensional high-pressure oedometer, (**b**) direct shear apparatus, and (**c**) triaxial shear instrument.

**Figure 6 materials-17-05277-f006:**
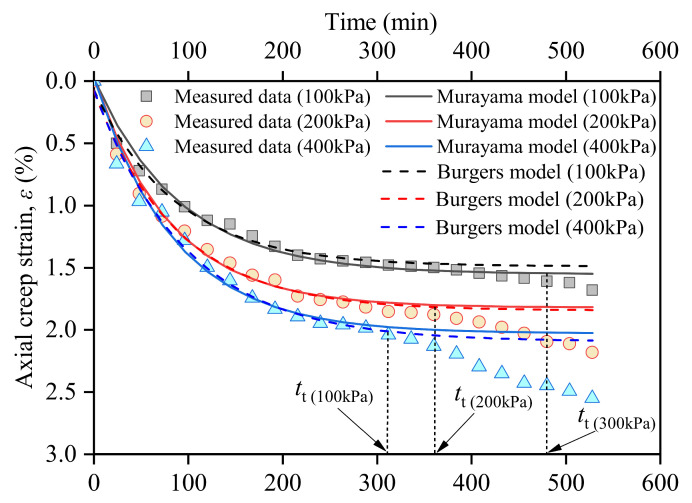
Relationship between axial compressive strain and elapsed time.

**Figure 7 materials-17-05277-f007:**
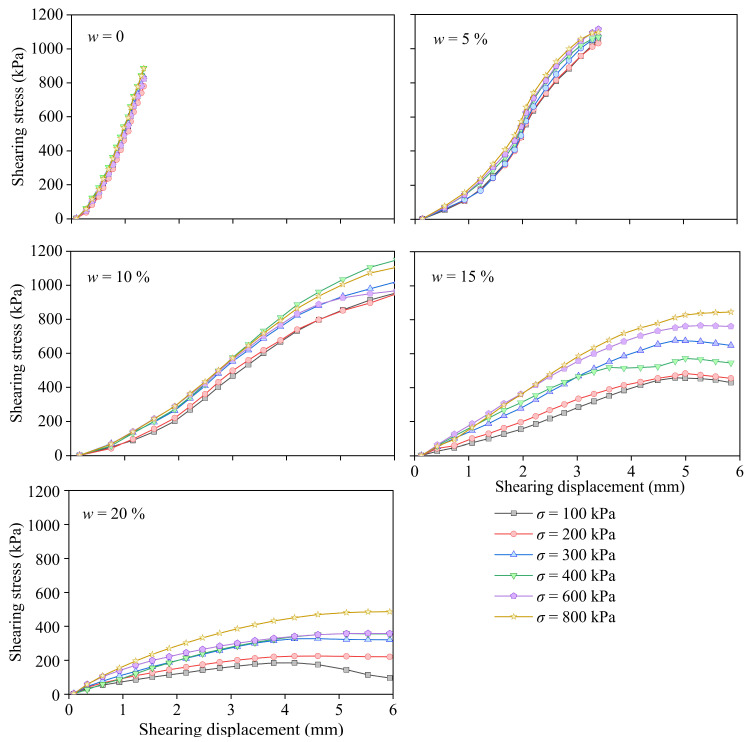
Direct shear results of calcareous sand after grouting under conditions of different moisture contents.

**Figure 8 materials-17-05277-f008:**
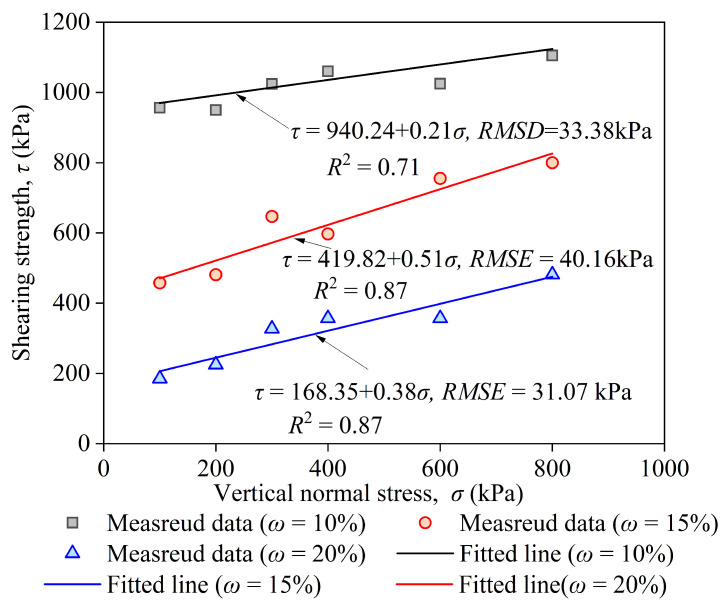
Molar strength envelope lines of the calcareous sand with different moisture content after grouting.

**Figure 9 materials-17-05277-f009:**
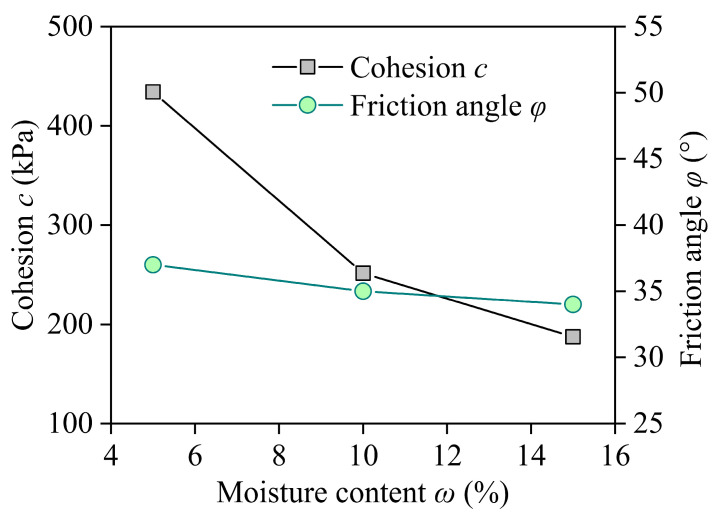
Internal friction angle φ and cohesive force *c* change with moisture content ω.

**Figure 10 materials-17-05277-f010:**
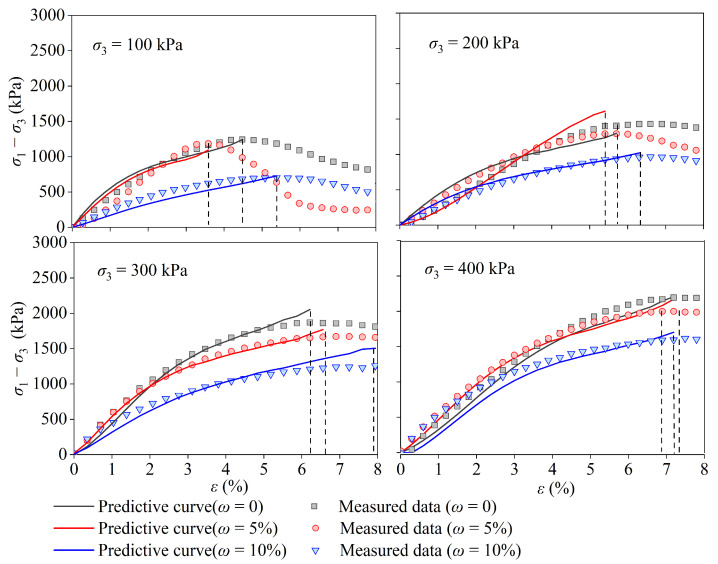
Predicted and measured stress–strain curves of coral calcareous sand after grouting.

**Figure 11 materials-17-05277-f011:**
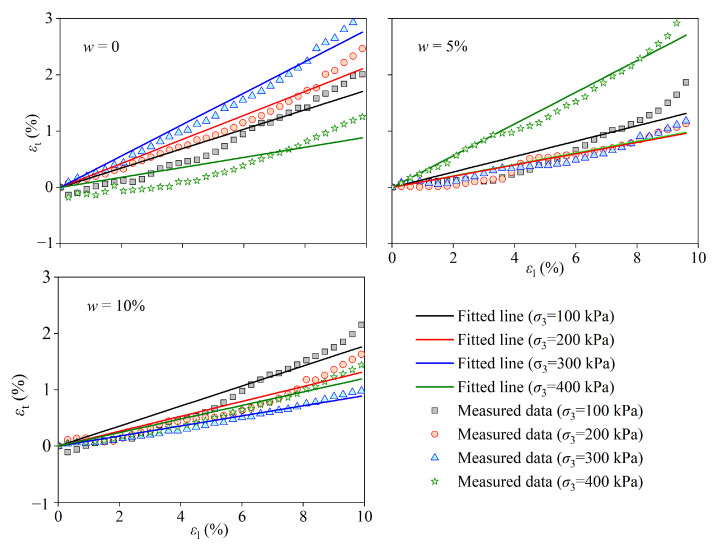
Relationship between transverse and longitudinal strain diagrams of coral calcareous sand with different water contents improved by polyurethane high polymer.

**Figure 12 materials-17-05277-f012:**
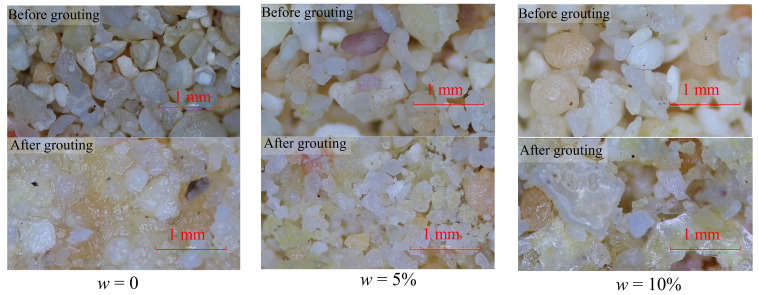
Optical microscope photo of the calcareous sand reinforced by PPA.

**Figure 13 materials-17-05277-f013:**
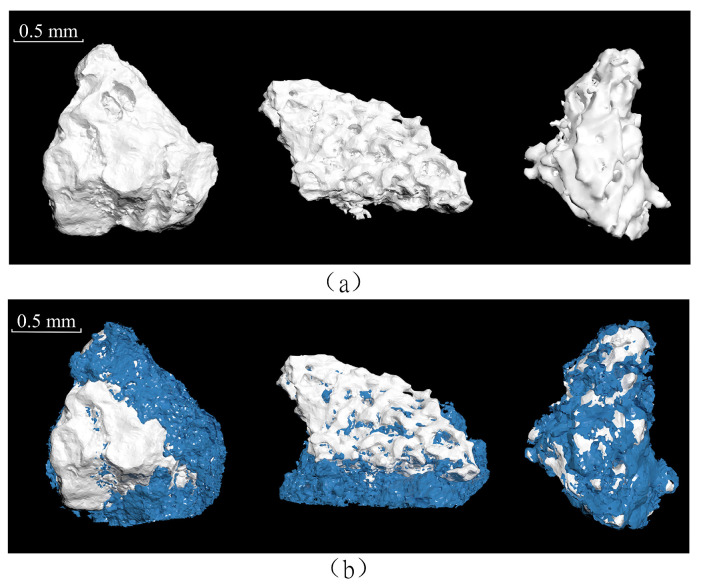
CT scanned images of calcareous sand particles and surrounding wrapped PPA: (**a**) discrete coral sand particles, and (**b**) coral sand particles connected by PPA (gray represents calcareous sand particles, and blue represents PPA).

**Figure 14 materials-17-05277-f014:**
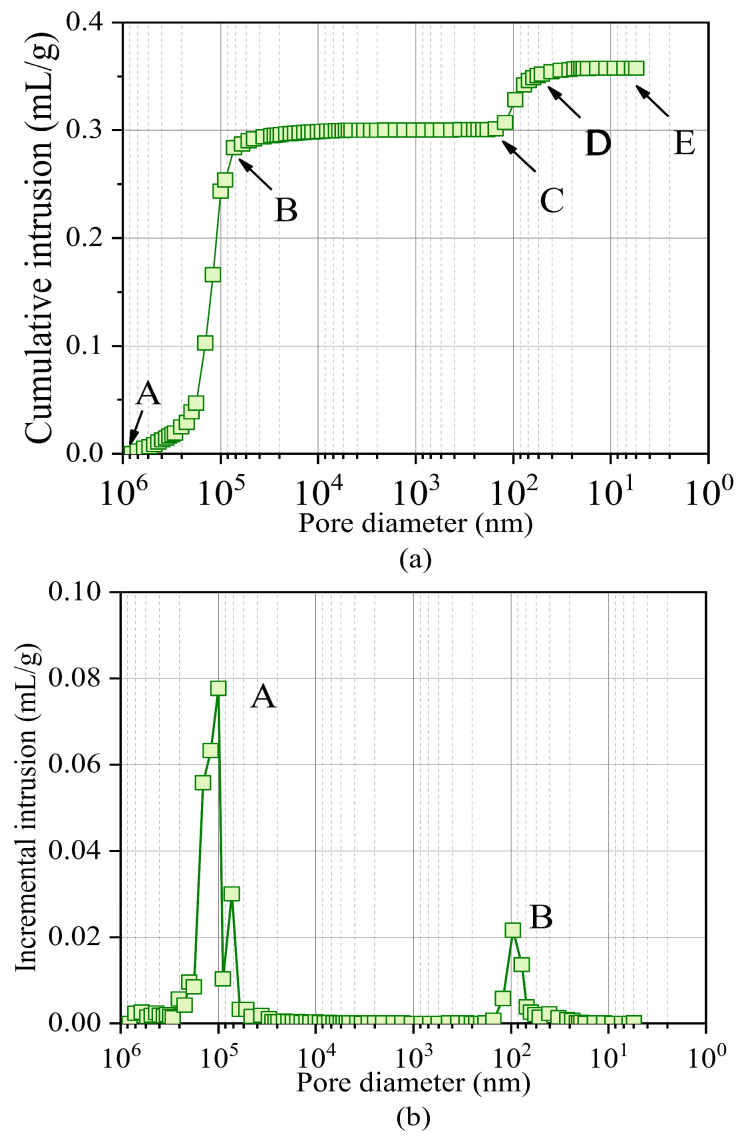
Relationship between cumulative intrusion volume, incremental intrusion, and pore diameter: (**a**) relationship between pore diameter and cumulative intrusion, and (**b**) and relationship between pore diameter and incremental intrusion.

**Table 1 materials-17-05277-t001:** Chemical components of coral calcareous sand obtained by X-Ray Fluorescence (XRF).

Element	Na	Mg	Al	Si	P	S	Cl	K	Ca	Fe	Sr
Percentage (%)	0.666	2.556	0.037	0.223	0.037	0.378	0.260	0.031	93.888	0.057	1.867

**Table 2 materials-17-05277-t002:** Chemical components of the permeable polyurethane polymer adhesive.

Ingredient Name		Molecular Formula	Density g/mL	Content	Features
A	Polymeric MDI	C_15_H_10_N_2_O_2_	1.2	50–70%	Brown liquid with a pungent odor
	MDI	C_15_H_10_N_2_O_2_	1.19	30–50%	White or yellow flakes or crystals
B	Polyether polyols	C_8_H_22_O_7_	1.02	30–60%	Light yellow transparent liquid
	Tris Nphosphate	C_9_H_18_Cl_3_O_4_P	Null	10–40%	Transparent, colorless, viscous

**Table 3 materials-17-05277-t003:** Basic parameters of the tests.

Tests	Moisture (%)	$\sigma_{1}$ (kPa)	$\sigma_{3}$ (kPa)	Dry Density (g/cm^3^)
Compression	10	100, 200, 400	0	1.45
Direct shear	0, 5, 10, 15, 20	100, 200, 300, 400, 600, 800	0	1.45
Triaxial shear	0, 5, 10	/	100, 200, 300, 400	1.45

**Table 4 materials-17-05277-t004:** *R*^2^ and *RMSD* of the Burgers and Murayama models.

Models	100 kPa		200 kPa		400 kPa	
	*R* ^2^	*RMSD*	*R* ^2^	*RMSD*	*R* ^2^	*RMSD*
Burgers	0.981	0.071	0.961	0.128	0.944	0.221
Murayama	0.987	0.559	0.989	0.137	0.986	0.193

**Table 5 materials-17-05277-t005:** *R*^2^ and *RMSD* of the fitted results of the Duncan–Chang model.

σ_3_ (kPa)	*w* = 5%		*w* = 10%		*w* = 15%	
	*R* ^2^	*RMSD* (kPa)	*R* ^2^	*RMSD* (kPa)	*R* ^2^	*RMSD* (kPa)
100	0.964	76.75	0.934	134.82	0.983	105.32
200	0.969	89.06	0.959	158.51	0.986	125.32
300	0.981	84.84	0.987	37.97	0.976	96.66
400	0.986	97.79	0.982	72.50	0.962	91.68

**Table 6 materials-17-05277-t006:** Poisson’s ratio and corresponding *R*^2^ of the calcareous sand reinforced by PPA.

σ_3_/kPa	*w* = 5%		*w* = 10		*w* = 15	
	*μ*	*R* ^2^	*μ*	*R* ^2^	*μ*	*R* ^2^
100	0.222	0.970	0.177	0.895	0.224	0.971
200	0.234	0.973	0.124	0.960	0.152	0.935
300	0.301	0.985	0.113	0.936	0.100	0.982
400	0.140	0.913	0.296	0.979	0.140	0.962

**Table 7 materials-17-05277-t007:** Pore diameter of calcareous sand reinforced by PPA.

Porosity (%)	Average Pore Diameter (nm)	Media Pore Diameter, V (mm)	Median Pore Diameter, A (nm)	Backbone Fractal Dimension	Percolation Fractal Dimension
37.88	474.95	1.16	80.74	3.00	2.98

## Data Availability

Data will be made available on request.

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
