# Peer review of "Investigation of the Mechanical Properties of Reinforced Calcareous Sand Using a Permeable Polyurethane Polymer Adhesive"

_materials, 2024, doi:10.3390/ma17215277_

Round 1
Reviewer 1 Report
Comments and Suggestions for Authors
The manuscript titled “Investigation of the mechanical properties of reinforced calcareous sand using a permeable polyurethane polymer adhesive” by Cao, D.; et al. is a scientific work where the authors assessed the mechanical properties of calcareous sand samples reinforced with polyurethane for direct shear and triaxial measurements, respectively. The most relevant outcomes found in this research could pave the way in the design of the next-generation of materials for building construction. This is a topic of growing importance and the manuscript is generally well-written. However, it exists some points that need to be addressed (please, see them below detailed point-by-point) to improve the scientific quality of the submitted manuscript paper before this article will be consider for its publication in Materials.
1) The authors should consider to add the term “mechanical properties” in the keyword list.
2) “Coral calcareous sand (…) become the main building material (…) and marine underground structures” (lines 36-41). Could the authors provide quantitatitve data insights according to the worldwide economic impact of the use of coral calcareous sand in the building Industrial sector? This will aid the potential readers to better understand the significance of this devoted research.
3) “When A and B are mixed (…) Under the action of a trimerization catalyst, isocyanate forms a six-membered heterocyclic structure composed of carbon and nitrogen atoms called isocyanurate” (lines 131-142). What was the chemical yield of this reaction? Some information needs to be furnished in this regard.
4) Figure 6 (line 272). The standard deviation bars for all the examined conditions should be furnished. Same comment for the Fig. 7 (line 318), the Fig. 8 (line 334) and the Fig. 9 (387).
5) Then, the regression coefficient of all the fittings should be also stated.
6) Fig. 11 (line 429). Why did the authors fit the transverse and longitudinal strain experimental ratio data with a linear regression? They should try a hyperbolic trend to improve the fitting.
7) “3.4. Microstructural mechanism analysis of the PPA reinforced calcareous sand. (…) When w was 0, the pores were evently distributed before grouting and almost all the pores were filled with PPA (…) When w = 5% (….) Many small pores, which were occupied by water before grouting were evenly distributed after grouting and oven drying (…) interparticle pore volume” (lines 434-446). Here, even if I agree with this statement provided by the authors it should be also mentioned the importance of the variation of local nanomechanics [1] on the coral sand and how they affect the permeability coefficient being increased when the pore ratio increases [2].
[1] Magazzù, A.; et al. Investigation of Soft Matter Nanomechanics by Atomic Force Microscopy and Optical Tweezers: A Comprehensive Review. Nanomaterials 2023, 13, 963. https://doi.org/10.3390/nano13060963
[2] Lv, C.; et al. Experimental Study on the Mechanical Strength, Deformation Behavior and Infiltration Characteristics of Coral Sand. Sustainability 2024, 16, 3479. https://doi.org/10.3390/su16083479
8) Conclusions (lines 495-532). This section perfectly remarks the most relevant outcomes found by the authors in this work and also the promising future perspectives. It may be opportune to add a brief statement to discuss about the potential future action lines to pursue the topic covered in this research.
Author Response
Comments 1: [The manuscript titled “Investigation of the mechanical properties of reinforced calcareous sand using a permeable polyurethane polymer adhesive” by Cao, D.; et al. is a scientific work where the authors assessed the mechanical properties of calcareous sand samples reinforced with polyurethane for direct shear and triaxial measurements, respectively. The most relevant outcomes found in this research could pave the way in the design of the next-generation of materials for building construction. This is a topic of growing importance and the manuscript is generally well-written. However, it exists some points that need to be addressed (please, see them below detailed point-by-point) to improve the scientific quality of the submitted manuscript paper before this article will be consider for its publication in Materials.
The authors should consider to add the term “mechanical properties” in the keyword list.]
Response 1: [Thanks for your positive and constructive comments. The manuscript has been revised point by point according to your suggestions. The new term “mechanical properties” has been added in the revised manuscript.]
Comments 2: [“Coral calcareous sand (…) become the main building material (…) and marine underground structures” (lines 36-41). Could the authors provide quantitatitve data insights according to the worldwide economic impact of the use of coral calcareous sand in the building Industrial sector? This will aid the potential readers to better understand the significance of this devoted research.]
Response 2: [Thanks for your suggestion. We have not found the macroeconomic data on the cost savings of calcareous sand. We have added a citation about ship cost from mainland. Please refer lines 39-43. Lines 39-43:“Because of the high transportation cost of traditional quartz river sand, CCS has become the main building material for reclamation projects such as runway foundations, offshore platforms, and marine underground structures. The carbon emission and transportation energy consumptions can reach 0.009 kg and 0.13 MJ when one ton of raw construction materials is shipped per kilometer [4]”]
Comments 3: [“When A and B are mixed (…) Under the action of a trimerization catalyst, isocyanate forms a six-membered heterocyclic structure composed of carbon and nitrogen atoms called isocyanurate” (lines 131-142). What was the chemical yield of this reaction? Some information needs to be furnished in this regard.]
Response 3: [Thanks for your suggestion. The chemical yield has been added. Please refer line 134. Line 134:“The chemical yields mainly include carbanmate and isocyanurate”]
Comments 4: [Figure 6 (line 272). The standard deviation bars for all the examined conditions should be furnished. Same comment for the Fig. 7 (line 318), the Fig. 8 (line 334) and the Fig. 9 (387).]
Response 4: [Thanks for your suggestions! These figures have been corrected in the revised manuscript.
For Fig.6, please refer lines 274-275, and revised Table 4. Lines 274-275: The root mean square deviations (RMSD) and R2 of Murayama and burgers models are listed in Table 4.
Table 4 R2 and RMSD of the Burgers and Murayama models
Models |
100 kPa |
200 kPa |
400 kPa |
|||
R2 |
RMSD |
R2 |
RMSD |
R2 |
RMSD |
|
Burgers |
0.981 |
0.071 |
0.961 |
0.128 |
0.944 |
0.221 |
Murayama |
0.987 |
0.559 |
0.989 |
0.137 |
0.986 |
0.193 |
Fig.7 presents the measured data without conducting parallel tests, so there is no deviation.
Fig.8 was redrawn and the RMSD was written on it.
Fig. 8. Molar strength envelope lines of the calcareous sand with different moisture content after grouting.
Fig.9 resents the measured data without conducting parallel tests, so there is no deviation.]
Comments 5: [Then, the regression coefficient of all the fittings should be also stated.]
Response 5: [Thanks for your kind reminder. All the regression coefficients for each fitting have been added.
Comments 6: [Fig. 11 (line 429). Why did the authors fit the transverse and longitudinal strain experimental ratio data with a linear regression? They should try a hyperbolic trend to improve the fitting.]
Response 6: [It is an interesting idea to use a hyperbolic function. However, we must use the liner fitting because the Poisson ration is a very common and important parameter in material field, which is defined as the ratio of transverse contraction (or expansion) strain to longitudinal extension strain in the direction of stretching force. Therefore, it must be a straight line through the origin according to its definition. This concept was proposed by the great scientist, Simeon Denis Poisson, in 1829.]
Comments 7: [“3.4. Microstructural mechanism analysis of the PPA reinforced calcareous sand. (…) When w was 0, the pores were evently distributed before grouting and almost all the pores were filled with PPA (…) When w = 5% (….) Many small pores, which were occupied by water before grouting were evenly distributed after grouting and oven drying (…) interparticle pore volume” (lines 434-446). Here, even if I agree with this statement provided by the authors it should be also mentioned the importance of the variation of local nanomechanics [1] on the coral sand and how they affect the permeability coefficient being increased when the pore ratio increases [2].]
[1] Magazzù, A.; et al. Investigation of Soft Matter Nanomechanics by Atomic Force Microscopy and Optical Tweezers: A Comprehensive Review. Nanomaterials 2023, 13, 963. https://doi.org/10.3390/nano13060963
[2] Lv, C.; et al. Experimental Study on the Mechanical Strength, Deformation Behavior and Infiltration Characteristics of Coral Sand. Sustainability 2024, 16, 3479. https://doi.org/10.3390/su16083479
Response 7: [Thanks for your recommendation. We have carefully read the papers you recommended and supplemented this section. The nanomechanics on the coral sand and how they affect the permeability coefficient have been explained. These two papers have been cited in the revised manuscript. Please refer lines 466-480.
Lines 466-480: “The Young’s modulus of CCS without grouting can be described by Hertz model which does not account for adhesion forces. Two assumptions are proposed for this model. The first assumption is that the nanoindentor is a perfect sphere that causes a perpendicular penetration to a perfectly planar surface, and the second assumption is that the strain-tress relation is linear with satisfying the Hooke’s law. Thus the Young’s modulus (E) can be expressed as [47]:
(18)
Where F is the load force (kN), R is the sphere radius of the nanoindenter (mm), and a is the contact radius between both surfaces. After grouting PPT, strong adhesive force exists among particles, and the Johnson, Kendall and Robers (JKR) model which considering the short-range forces between surface contact area and external sample surface [47]. The JKR model is used for the occasion with large spherical indenters exposing larger contact areas with larger adhesion events, which can be described as [47]:
(19)
Where is the required energy to separate a unitary area of both surfaces (J).”]
Comments 8: [Conclusions (lines 495-532). This section perfectly remarks the most relevant outcomes found by the authors in this work and also the promising future perspectives. It may be opportune to add a brief statement to discuss about the potential future action lines to pursue the topic covered in this research.]
Response 8: [Thanks for your suggestion. A new paragraph has been added to introduce the potential application in the future. Please refer to lines 556-561.
Lines 556-561:“In actual engineering applications, further research on grouting methods is needed when using PPA to reinforce calcareous sand, such as the relationship between grouting pressure and solidification time, effective grouting radius, PPA-CCS mixed body morphology, and grouting pipe layout spacing. When the slurry solidifies too slowly, it may flow along the water flow channel; while when the slurry solidifies too quickly, it may be difficult to form an intact solid, resulting in failure in foundation treatment. ”]

Reviewer 2 Report
Comments and Suggestions for Authors
In this study, a permeable PPA (Polyurethane Polymer Adhesive) was used to reinforce calcareous sand, and a new grouting equipment was designed to prepare soil samples with PPA for geotechnical tests to replace the traditional mixture-stirring method. The compressive and shearing performances were evaluated using one-dimensional direct shear and triaxial shear tests.
The idea developed by the Authors undoubtedly has important implications from the point of view of possible engineering applications. However, the paper requires several changes and additions.
Lines 118-119
“The effective grain size (d10), d30,”
Check the position of d10 and d30 in this sentence. Should they both be included in parentheses?
Line 129
“The PPA used in this study was a two-component (referred to as A and B)”
Authors should provide information on the two-components that they will indicate as A and B.
Lines 131-132
“When A and B are mixed, two main chemical reactions occur”
Authors should specify whether the two chemical reactions occur independently (contemporarily) or whether the second requires the first to have already provided its final products in order to occur.
Line 140
“Under the action of a trimerization catalyst”
Authors should provide information on the catalyst.
Fig 2(a)
Check the spelling of the groups.
Lines 277-278
“E0 and E1 is the elastic modulus (MPa)”
Since the elastic moduli in the formula are actually two (E0 and E1), the Authors should specify the difference between them.
Line 278
“η1 and η2 is the viscosity coefficient (MPas)”
Authors should specify the difference between the two viscosity coefficients.
Authors should check the unit of measurement of the viscosity coefficient: they should separate Pa (Pascal) from s (second).
Line 320
“Under higher moisture content conditions ( )”
Is there any missing text in brackets?
Line 371
“Ducan and Chang suggested using LS = 70% and LS =95%”
Is the first scholar Duncan instead of Ducan?
Lines 375-377
“The cohesive force c and internal friction angle 𝜑 for different moisture contents were obtained by plotting the Mohr stress circle and shear strength envelopes, as shown in Fig. 9.”
The Authors already used the symbol 𝜑 to indicate the interface friction angle, which is conceptually different from the internal friction angle. Therefore, two different symbols should be used to indicate the two friction angles, if the Authors are actually talking about two different things.
Fig. 9 does not show the Mohr stress circles and their envelopes. Therefore, Fig. 9 should not be cited in this sentence. It could be cited in the following sentence:
The cohesive force c decreased with increasing moisture content (Fig. 9) …
Lines 394-397
“As shown in Fig. 10, the sand samples reinforced by PPA exhibited stress peaks and obvious strain-softening characteristics under low confining pressures (σ3 = 100 or 200 kPa), whereas those under high confining pressure conditions (400 kPa) exhibited strain-hardening features.”
This seems in contradiction to contradict what was stated two sentences earlier, since sand samples subjected to low confining pressures should be in a looser state than sand samples subjected to high confining pressure. Therefore, based on what was stated two sentences earlier, they should exhibit strain-hardening characteristics, whereas samples subjected to high confining pressure should exhibit strain-softening characteristics.
Lines 424-426
“It was reported that µ decreased with an increase in the magnitude of the confining pressure and relative density before softening, which is consistent with the observations in Fig. 11”
It seem consistent for w=10% but not for other percentages of w.
Comments on the Quality of English Language
Minor editing of English language required.
Author Response
Comments 1: [In this study, a permeable PPA (Polyurethane Polymer Adhesive) was used to reinforce calcareous sand, and a new grouting equipment was designed to prepare soil samples with PPA for geotechnical tests to replace the traditional mixture-stirring method. The compressive and shearing performances were evaluated using one-dimensional direct shear and triaxial shear tests.
The idea developed by the Authors undoubtedly has important implications from the point of view of possible engineering applications. However, the paper requires several changes and additions.
Response 1: [Thank you very much for your positive comments. We have revised the manuscript point by point according to your reviews.]
Comments 2: Lines 118-119: “The effective grain size (d10), d30,” Check the position of d10 and d30 in this sentence. Should they both be included in parentheses?]
Response 2: [ We are sorry for making such a mistake. We have corrected it. Please refer lines 120-122. Lines 120-122:“The effective grain size (d10), median grain size (d30), and control grain size (d60) were 0.044, 0.102, and 0.280 mm, respectively.”]
Comments 3: [Line 129: “The PPA used in this study was a two-component (referred to as A and B)” Authors should provide information on the two-components that they will indicate as A and B.]
Response 3: [Thanks for your suggestion. Details of A and B have been added in the revised manuscript. Please refer lines 134-136.
Lines 134-136:“The main composition of A is diphenylmethane diisocyanate. The compositions of B include polyether or polyester polyols, tris(1-chloro-ethylpropyl) phosphate, and additive.”]
Comments 4: [Lines 131-132: “When A and B are mixed, two main chemical reactions occur” Authors should specify whether the two chemical reactions occur independently (contemporarily) or whether the second requires the first to have already provided its final products in order to occur.]
Response 4: [These two reactions are independent. This ambiguous state has been corrected in the revised manuscript. Please refer to lines 136-137.
Lines 136-137: “When A and B are mixed, two main separate chemical independent reactions occur, as shown in Fig. 2.”]
Comments 5: Line 140: [“Under the action of a trimerization catalyst” Authors should provide information on the catalyst.]
Response 5: [Thanks for your suggestions. We have supplemented it. Please refer lines 150-155.
Lines 150-155:“Some organic metal compounds and nitrogen element compounds can be used as trimerization catalysts for isocyanates. The commonly used tertiary amine trimeriza-tion catalysts include N-N-dimethylcyclohexylamine, and N-N'-diethylpiperazine. When manufacturing isocyanurate, in order to avoid the formation of useless sub-stances with excessive polymerization due to unlimited reaction, a small amount of inhibitor is generally added to control the degree of trimerization such as phosphoric acid, dimethyl sulfate, etc.”]
Comments 6: [Fig 2(a): Check the spelling of the groups.]
Response 6: [This word has been deleted in the revised manuscript.]
Comments 7: [Lines 277-278: “E0 and E1 is the elastic modulus (MPa)” Since the elastic moduli in the formula are actually two (E0 and E1), the Authors should specify the difference between them.]
Response 7: [Thanks for pointing out the problems. We have corrected them. Please refer lines 290-292.
Lines 290-292: “ E0 is the elastic modulus of Hooke body (MPa); and E1 is the elastic modulus of Kelvin body (MPa)”].
Comments 8: [Line 278: “η1 and η2 is the viscosity coefficient (MPas)” Authors should specify the difference between the two viscosity coefficients. Authors should check the unit of measurement of the viscosity coefficient: they should separate Pa (Pascal) from s (second).]
Response 8: [Thanks for pointing out the problems. We have corrected them. Please refer lines 292-293.
Lines 292-293: “η1 is the viscosity coefficient in the branch path (kPa), and η2 is the viscosity coeffi-cient in the branch path (kPa).”]
Comments 9: [Line 320: “Under higher moisture content conditions ( )” Is there any missing text in brackets?]
Response 9: [Sorry for making such a mistake. We have corrected it. Please refer lines 336-337.
Lines 336-337: “Under higher moisture content conditions (w > 10%), the shear strength increased with an increase in the vertical normal stress.”]
Comments 10: [Line 371: “Ducan and Chang suggested using LS = 70% and LS =95%” Is the first scholar Duncan instead of Ducan?]
Response 10: [Yes, it should be Duncan. We have corrected the spelling.]
Comments 11: [Lines 375-377: “The cohesive force c and internal friction angle φ for different moisture contents were obtained by plotting the Mohr stress circle and shear strength envelopes, as shown in Fig. 9.”
The Authors already used the symbol φ to indicate the interface friction angle, which is conceptually different from the internal friction angle. Therefore, two different symbols should be used to indicate the two friction angles, if the Authors are actually talking about two different things.
Fig. 9 does not show the Mohr stress circles and their envelopes. Therefore, Fig. 9 should not be cited in this sentence. It could be cited in the following sentence:
The cohesive force c decreased with increasing moisture content (Fig. 9)] …
Response 11: [Many thanks for your careful checking. After consulting the original literature, we found the use of “interface friction angle” is wrong. We have corrected it as internal friction angle. The citation location has been moved to the right place as well. Please refer to lines 391-394.
Lines 391-394: “The cohesive force c and internal friction angle φ for different moisture contents were obtained by plotting the Mohr stress circle and shear strength envelopes. The cohesive force c decreased with increasing moisture content because the polyurethane polymer strengthened the soil particle skeleton and filled the internal pores, as shown in Fig.9.”]
Comments 12: [Lines 394-397: “As shown in Fig. 10, the sand samples reinforced by PPA exhibited stress peaks and obvious strain-softening characteristics under low confining pressures (σ3 = 100 or 200 kPa), whereas those under high confining pressure conditions (400 kPa) exhibited strain-hardening features.”
This seems in contradiction to contradict what was stated two sentences earlier, since sand samples subjected to low confining pressures should be in a looser state than sand samples subjected to high confining pressure. Therefore, based on what was stated two sentences earlier, they should exhibit strain-hardening characteristics, whereas samples subjected to high confining pressure should exhibit strain-softening characteristics.]
Response 12: [Thanks for your careful observation. We have changed the statement. Please refer to lines 407-410.
Lines 407-410:“When calcareous sand is in a loose state, the stress-strain relationship exhibits strain-softening characteristics, whereas in a dense state, it exhibits strain-hardening characteristics.”]
Comments 13: [Lines 424-426: “It was reported that µ decreased with an increase in the magnitude of the confining pressure and relative density before softening, which is consistent with the observations in Fig. 11” It seem consistent for w=10% but not for other percentages of w.]
Response 13: [This sentence has been deleted in the revised manuscript.]

Round 2
Reviewer 1 Report
Comments and Suggestions for Authors
The authors did a great deal of effort to cover all the suggestions raised by the Reviewers. For this reason, the scientific manuscript quality was greatly improved. Based on the signifance and novelty of the gathered results, I warmly endorse this work for further publication in Materials.
Author Response
Comments 1: [Lines 294-295“η1 is the viscosity coefficient in the branch path (kPa), and η2 is the viscosity coefficient in the branch path (kPa).”]
Response 1: [We have corrected it. is the viscosity coefficient of Hooke body (kPa), and is the viscosity coefficient of Kelvin (kPa).]
Comments 2: [Minor editing of English language required]
Response 2: [Thanks for your suggestion. We have carefully checked through the draft and re-written the language]
Reviewer 2 Report
Comments and Suggestions for Authors
The authors reviewed the manuscript, but there was probably a copy-paste problem.
Lines 294-295
“η1 is the viscosity coefficient in the branch path (kPa), and η2 is the viscosity coefficient in the branch path (kPa).”
Comments on the Quality of English Language
Minor editing of English language required.
Author Response

(The authors gave the same response as above.)
